# Effectiveness of Group CBT on Internalizing and Externalizing Symptoms in Children with Mixed Psychiatric Disorders

**DOI:** 10.3390/children9111602

**Published:** 2022-10-22

**Authors:** Sarianna Barron-Linnankoski, Hanna Raaska, Paula Bergman, Eija Närvänen, Marko Elovainio, Marja Laasonen

**Affiliations:** 1Department of Child Psychiatry, Helsinki University Hospital and University of Helsinki, P.O. Box 827, 00029 Helsinki, Finland; 2The Social Insurance Institution of Finland, P.O. Box 450, 00056 Helsinki, Finland; 3Biostatistics Consulting, Department of Public Health, University of Helsinki and Helsinki University Hospital, 00014 Helsinki, Finland; 4Department of Psychology and Logopedics, University of Helsinki, 00014 Helsinki, Finland; 5Department of Psychology and Logopedics/Research Program Unit, University of Helsinki, 00014 Helsinki, Finland; 6The Finnish Institute for Health and Welfare, 00270 Helsinki, Finland; 7Logopedics, Philosophical Faculty, School of Humanities, University of Eastern Finland, 80100 Joensuu, Finland

**Keywords:** cognitive behavioral therapy, group cognitive behavioral therapy, internalizing symptoms, externalizing symptoms, children

## Abstract

Background: Our study addressed the gap in research on the effectiveness of cognitive behavioral therapy (CBT) in treating children with mixed psychiatric disorders. We examined the immediate and long-term effects of group CBT (GCBT), delivered in naturalistic clinical settings, on reducing internalizing and externalizing symptoms in children with mixed psychiatric disorders. Further, we compared the effectiveness of cost-effective, manualized GCBT to treatment as usual (TAU) consisting of individually tailored psychiatric outpatient services delivered by mental health care specialists. Methods: Children aged 6–12 years (*n* = 103) diagnosed with psychiatric disorders, more than 70% with psychiatric comorbidity, were assigned either directly to GCBT (GCBT group; *n* = 52) or TAU for approximately 3 months, after which they received GCBT (TAU + GCBT group; *n* = 51). Internalizing and externalizing symptoms were assessed using parent- and teacher-report questionnaires (Child Behavior Checklist and Teacher Report Form) at referral to treatment, pre-treatment, post-treatment, and six-month follow-up. Results: Parent- and teacher-rated internalizing symptoms and parent-rated externalizing symptoms were reduced immediately after GCBT. Long-term GCBT gains were prominent for parent-rated externalizing symptoms. No differences were observed between the effectiveness of GCBT and TAU. Conclusions: Our results suggest that GCBT and TAU services are equally effective in treating internalizing and externalizing symptoms in children with mixed psychiatric disorders, providing support for the broader use of cost-effective manualized GCBT. Manualized GCBT, which requires relatively short training, can also be delivered at primary healthcare levels. Our results are of relevance to cost-effectiveness and global mental health staff shortages.

## 1. Introduction

Cognitive behavioral therapy (CBT) has been extensively researched in children with internalizing disorders and has been shown effective in the treatment of anxiety symptoms in numerous systematic reviews and meta-analyses [1,2,3,4,5,6]. While the evidence is mainly based on highly controlled conditions in research settings, less is known of CBT effectiveness when delivered as part of routine treatment in clinical settings. However, the findings of a recent systematic review/meta-analysis supported the effectiveness of CBT conducted in routine clinical care for the treatment of internalizing symptoms in children with anxiety, depression, post-traumatic stress disorder, and obsessive–compulsive disorder, with outcomes comparable to those found in university settings [7].

The effectiveness of CBT in the treatment of anxiety has been demonstrated in both individual and group formats [2,3,4,5,8], using disorder-specific and generic protocols [5], and using self and parent reports [4,9]. Furthermore, youth between the ages of 6 and 19 years have been suggested to benefit on the same level [10]. The use of multiple informants in the assessment of psychopathology is warranted [11], but so far, to the present authors’ knowledge, research using teachers as informants is lacking regarding CBT delivered in treatment settings. Teachers interact with school-aged children on an almost daily basis, and their evaluations offer further insights into the children’s behavior in terms of context and perspective [12]. Evidence of the durability of CBT effects in the treatment of internalizing disorders is limited [2,3,5], but there are preliminary findings of gains maintained or increased at ca. 6–10 month follow-up, e.g., [3,7].

Several meta-analyses have shown CBT also to be effective in reducing externalizing symptoms such as anger-related [13], aggressive [14], and antisocial behavior problems in children [15]. In addition, CBT has been found effective in the treatment of externalizing symptoms in childhood attention deficit hyperactivity disorder (ADHD) and oppositional defiant disorder (ODD) [16]. However, to our knowledge, only one systematic review/meta-analysis has focused primarily on the effectiveness of CBT in the routine clinical treatment of children with externalizing disorders (ADHD, conduct disorder, and ODD) and has shown treatment effects in naturalistic settings comparable to those in university settings [17]. The effectiveness of CBT in children with externalizing disorders has been shown in individual and group formats [17] and based on self-, parent, and teacher ratings, e.g., [13,14,15,16]. Younger age has been associated with larger symptom reductions after CBT [17]. Preliminary evidence suggests that the effects of CBT in the treatment of children with behavioral problems are maintained at ca. 4–10-month follow-up, e.g., [15,17,18].

There is a gap in the research on CBT in the treatment of children with a wider range of psychiatric disorders exhibiting both internalizing and externalizing symptoms. Comorbidities have been shown to be common within and between internalizing and externalizing groupings [19,20,21,22]. Nevertheless, the existing research on transdiagnostic CBT has focused on children with internalizing disorders [8,23]. There is, however, preliminary support for the use of CBT in clinical child populations with a range of psychiatric symptomology. Anxiety-focused CBT has been suggested to reduce comorbid externalizing symptoms in childhood anxiety disorders [24,25], and CBT delivered to children with externalizing disorders has been associated with decreases in comorbid internalizing symptoms [16]. Further research is needed on interventions that can be implemented at an early stage in transdiagnostic pediatric populations [26]. The conceptualization of psychopathology has, on the other hand, shifted from a categorical, diagnostic approach towards a more dimensional approach that reflects individual differences quantitatively as graded continuums, facilitating issues of heterogeneity and comorbidity, e.g., [27,28]. The dimensional classification includes the symptoms and signs of the categorical diagnostic classification, allowing broader monitoring of behaviors for changes in severity [29]. Children with symptoms on both the internalizing and externalizing dimensions of behavior are a common but so far understudied population in clinical care settings.

According to the available systematic reviews/meta-analyses there is little evidence that CBT is superior to treatment as usual (TAU) in the treatment of internalizing symptoms [2,3,30,31]. The evidence is based on studies comparing interventions using CBT-specific techniques (e.g., psychoeducation, cognitive problem solving, relaxation training, and exposure) and TAU methods that may utilize a wider range of non-CBT psychosocial or pharmacological treatments. Although GCBT has been shown effective but not superior to TAU [31], the potential benefits of GCBT over TAU may still be seen as substantial given the current severe global shortage of mental health professionals and financial resources [32]. Considering the evidence of increases in children´s internalizing and externalizing problems during recent years [33], research comparing intervention effects in children with multi-dimensional psychiatric problems is warranted so that clinical treatment practices may be revised as necessary and the limited financial and staff resources allocated appropriately.

Our study addresses the gap in research on the effectiveness of GCBT delivered in naturalistic, real-world outpatient settings in treating children with mixed psychiatric disorders. We examined the effectiveness of GCBT in reducing parent- and teacher-rated internalizing and externalizing symptoms in a clinically referred sample of children with a variety of psychiatric disorders, with symptoms in both the internalizing and externalizing dimensions. In addition to assessing immediate effects following GCBT, we also examined whether the potential effects would be maintained at a 6-month follow-up. Furthermore, we compared the effectiveness of manualized GCBT to TAU, during which participants received routine care services tailored to each child individually while waiting for the onset of GCBT, to gain further insight into the potential utility of cost-effective GCBT. We used the Finnish versions of the CBT-based FRIENDS program [34,35,36], a generic treatment for anxiety disorders, as it was a protocol used at the Helsinki University Hospital (HUS) child psychiatric outpatient clinics (outpatient clinics) for children with a variety of psychiatric symptoms. The FRIENDS program follows the main principles of CBT, involving techniques and approaches that build an understanding of the link between our thoughts, feelings, and behavior. It focuses on reducing current symptomology and on promoting resilience and well-being. It has been shown to be effective in reducing children´s internalizing symptoms in several peer-reviewed studies using parent ratings, with improvements maintained or increased at a 12-month follow-up [37,38,39]. It was hypothesized that children with mixed psychiatric disorders would exhibit improvements in parent-rated internalizing symptoms and parent- and teacher-rated externalizing symptoms following the completion of the GCBT intervention. Based on prior data on internalizing symptoms [31], we hypothesized that GCBT would be at least as effective as TAU in treating children with a broad range of psychiatric symptoms.

## 2. Materials and Methods

### 2.1. Participants

Participants were recruited through the HUS outpatient clinic’s ordinary referral system between 2016 and 2018. Children aged 6–12 years are referred to these specialized psychiatric care outpatient clinics by general practitioners (e.g., at school health services, health centers, or family counseling centers), and all services are provided free of charge. Children were referred to GCBT if they exhibited symptoms of anxiety or depression, had deficiencies in emotional and behavioral skills impairing functioning, and possessed sufficient social and cognitive skills to participate in group work. Further, both children and their caregivers (parents) were to show motivation for the intervention. Exclusion criteria was excessive physical aggression and excessive physical restlessness. Referral and exclusion criteria were based on clinical and diagnostic assessments carried out by physician-led multidisciplinary teams (e.g., thorough clinical evaluation of the child, extensive data collection, and the use of diagnostic, structured methods; see 2.3. Measures Section). All children referred to GCBT in outpatient clinics were invited to the study. Children who were referred to GCBT but whose severity of psychiatric symptoms required inpatient treatment (e.g., acute suicidality) were not included in the study because the GCBT protocol was largely modified in inpatient units to better suit the treatment setting.

Following referral, children and their parents were asked to participate in the study. Of the 132 families invited to the study, a final sample of 103 children aged 6–12 years and their parents gave informed written consent for participation and use of measures, and were enrolled in the study. Further, informed written consent for the use of medical records in the collection of data on the children’s diagnoses, changes in medication, other ongoing treatments, schooling, and custody during the study was obtained for 94 children. Informed written consent for the use of medical records in the collection of data was not obtained for nine children. The flow diagram and study procedure are depicted in Figure 1 (see Section 2.5. Statistical Analysis Section for missing parent-/teacher-rated data at different time points). The baseline characteristics for participants are shown in Table 1. The present study is part of a larger treatment outcome study examining the effects of GCBT on the well-being of children with psychiatric disorders, approved by the HUS Ethics Committee for Women’s and Children’s Health and Psychiatry (49/13/03/03/2016 and HUS/2699/2018).

### 2.2. Procedure

After referral, the total sample of 103 children (the intent-to-treat ITT sample) was assigned either directly to GCBT (GCBT group, *n* = 52) or to a TAU waitlist condition after which they received GCBT (TAU + GCBT group, *n* = 51). Assignment to the GCBT or TAU + GCBT group was determined by the order in which children were referred to GCBT and the order in which GCBT groups were initiated in the age range in question. The mean duration of treatment for the GCBT condition was 4.7 months (SD = 1.0), consisting of 10 weekly group sessions and 2 booster sessions one and two months after session 10 (altogether 12 sessions), and 3.2 months (SD = 1.4) in the TAU condition consisting of routine outpatient care services tailored for each child individually. Children continuing HUS outpatient clinic care during the six-month follow-up condition (M = 6.33 months, SD = 1.29 months) also had access to TAU care services, if needed.

Parents and teachers completed questionnaires at four time points returning them either at an appointment or by post (teachers in all cases). Time 1 was at baseline; pre-GCBT for the GCBT group and pre-TAU for the TAU + GCBT group. Time 2 corresponded to post-GCBT for the GCBT group and post-TAU/pre-GCBT for the TAU + GCBT group. Time 3 was at a six-month follow-up for the GCBT group and post-TAU for the TAU + GCBT group, and Time 4 was at a six-month follow-up for the TAU + GCBT group (Figure 1).

### 2.3. Measures

The Child Behavior Check List (CBCL), part of the Achenbach System of Empirically Based Assessment [40], completed by parents, assesses broadband internalizing and externalizing symptoms in children aged 6 to 18 years. The CBCL comprises 113 items scored on a 3-point scale from 0 to 2 (0 = not true, 1 = somewhat/sometimes true, and 2 = very/often true). The CBCL contains eight empirically based syndrome scales: (1) anxious/depressed, (2) withdrawn/depressed, (3) somatic complaints, (4) social problems, (5) thought problems, (6) attention problems, (7) rule-breaking behavior, and (8) aggressive behavior. Three of these (withdrawn, somatic, and anxious/depressed) combine to form the internalizing score (CBCL-Int), two (rule-breaking and aggressive behavior) form the externalizing score (CBCL-Ext), and all eight form the total problem score (CBCL-Tot). Borderline T-scores for CBCL-Int, CBLC-Ext, and CBCL-Tot range from 60 to 63 (84th–90th percentile), with scores below 60 reflecting the normal range and scores above 63 the clinical range. The original version of CBCL has excellent internal consistency for CBCL-Int, CBCL-Ext, and CBCL-Tot; good to excellent internal consistency for the empirically based syndrome scales; and excellent test–retest reliability across the empirically based problem scales [40]. The Finnish version has also demonstrated excellent internal consistency for CBCL-Int and CBCL-Ext [41].

The Teacher’s Report Form TRF [40] is the teacher-rated counterpart of the CBCL, also comprising 113 items scored the same way as the CBCL. The original US version of the TRF has excellent internal consistency for the internalizing (TRF-Int), externalizing (TRF-Ext), and total problem scores (TRF-Tot); good to excellent internal consistency for the empirically based syndrome scales; and excellent mean test–retest reliability across the empirically based problem scales [40].

The participants´ gender, age, and mother’s educational level were requested in the questionnaires. Information regarding the participants´ diagnoses, potential medication, family type and custody, and potential child welfare notifications were collected from the medical reports of the 94 children for whom written consent was obtained. As all participants were referred to specialized psychiatric outpatient clinics, all diagnoses were conducted by specialists in child psychiatry according to diagnostic criteria of the 10th revision of the International Statistical Classification of Diseases and Related Health Problems ICD-10 [42]. The diagnostic processes were conducted as multidisciplinary evaluations led by M.D. specialists. The diagnostic processes included assessment of the child, extensive data collection from the child, parents, and school, and the use of required structured methods as appropriate (e.g., Attention Deficit/Hyperactivity Rating Scale [43]; Autism Diagnostic Observation Schedule, second edition [44]; Autism Diagnostic Interview, revised [45]). The information on teacher changes was obtained from the TRF.

### 2.4. Treatment

#### 2.4.1. The TAU Condition

The routine TAU outpatient services received by the TAU + GCBT group while waiting for the onset of GCBT were typically less intensive than the GCBT intervention. They consisted of individually tailored and individually provided care services, such as psychoeducation, supportive counseling, and help with family/school functioning provided by the child´s clinical care manager (nurse, psychologist, occupational therapist, or social worker). If additional support (i.e., speech, occupational, or family therapy sessions in a few cases) or changes in medication were needed, these were also provided.

#### 2.4.2. The GCBT Condition

Three versions of the FRIENDS program were used depending on the age range of the participants: the Finnish translations of Fun FRIENDS [35] for ages 6–8, FRIENDS for Life [36] for ages 9–12, and My FRIENDS [34] for those turning 13 during the study. The choice of the FRIENDS program as the CBT-based intervention was based on the treatment practices of HUS outpatient clinics, not on study-related factors. FRIENDS is an acronym symbolizing and helping children to remember the topics covered in the program: F = Feelings; R = Relaxation; I = Inner thoughts; E = Exploring strategies; N = Now reward yourself; D = Don´t forget to practice; S = Stay calm. The therapeutic techniques of FRIENDS include psychoeducation, empathy training and self-regulation, relaxation and mindfulness, cognitive strategies, exposure, social support training, self-rewards, and relapse prevention. These topics and techniques were addressed in a progressive and integrative way throughout the program. The versions of the FRIENDS program comprised ten weekly 60-min sessions, followed by two booster sessions approximately one and two months after session 10. The FRIENDS program also comprised two group sessions for parents, consisting mainly of psychoeducation.

A total of 31 groups were run at seven HUS outpatient clinics in the Helsinki metropolitan area, Finland. Children were divided into groups based on their age and order of referral. There were four to six children in each treatment group, and each group was delivered by two therapists (psychologists, occupational therapists, or nurses). In Finland, nursing education is a bachelor’s degree and is provided at universities of applied sciences. All therapists attended a one-day FRIENDS training workshop, as required by the FRIENDS program. Sessions were conducted as outlined in the treatment manual, with a focus on the core tasks depending on the children´s pace in completing exercises. However, there were a few groups (*n* = 6) lacking the second booster session, and some groups comprising only one parent group session (*n* = 6) or lacking parent group sessions (*n* = 2) due to practices of the clinics in question. Throughout the program, the children used the FRIENDS program workbooks [46,47,48] and were assigned homework, but its completion was not monitored. Parents were encouraged to help children with homework and to continue using the workbook after the intervention.

#### 2.4.3. Treatment Integrity

The study design was introduced and discussed at regular, periodic group leader meetings. Further, all therapists were contacted individually before the start of their group and kept closely informed of the study design via email and telephone. Therapists were encouraged to contact the research team for questions at any stage of the study. Therapists informed the research team of the number of child and parent sessions attended by each family taking part in the study.

### 2.5. Statistical Analysis

Total T-scores for the CBCL and TRF internalizing, externalizing, and total problems scores and syndrome scales were calculated using the Assessment Data Manager (ADM) version 9.1. Statistical analyses were performed using R version 4.0.3 R Development Core [49] and IBM SPSS Statistics version 25.0. All analyses were undertaken on an intent-to-treat (ITT) basis. All participants were retained in the group (GCBT or TAU + GCBT) they were assigned to at baseline, irrespective of adherence or attrition.

Z-tests were used to compare the proportion of missing data at T1, T2, and T3 between groups. Missing values (see Table 2) mostly originated from lacking or incomplete measures from parents or teachers, and to a far lesser degree from treatment dropouts. There were, for example, several cases where measures were returned by post, but perhaps by accident only one side of the two-sided form had been completed (parents *n* = 5, teachers *n* = 17), or parents had filled the TRF instead of the CBCL form (*n* = 6), and as a result, a total score could not be calculated. The GCBT and TAU + GCBT groups differed from each other only concerning missing TRF scores at T1 (*p* = 0.038). In addition to lacking measures, missingness at T1 was associated with the incompleteness of the two-sided form (*n* = 1 in the GCBT group; *n* = 3 in the TAU + GCBT group) and submission of forms during summer holidays (*n* = 6 in the TAU + GCBT group). No imputations were made because missingness was assumed to be completely at random.

Descriptive statistics are presented as means (M) and standard deviations (SD), medians (Mdn) and interquartile ranges (IQR = Q1–Q3), or frequencies and percentages (%), depending on the variable distributions. Outlying CBCL and TRF symptom scores and non-normal distributions were identified by graphically examining box plots. All outliers were determined as rare but legitimate values, and were, thus, included in the analyses.

We examined group differences in demographic data at baseline, the number of attended group sessions, and the discharge rate during follow-up using Chi-square tests or Fisher´s exact tests for nominal variables and Mann–Whitney U tests for continuous variables. In addition, z-tests were used to compare maternal education levels and parent session distributions between groups. As preliminary analyses, the changes in CBCL and TRF symptom scores over time (from entry to study at baseline to Time 4) among participants were analyzed using linear mixed models, and 95% confidence intervals (CIs) and *p* values were computed using the Wald´s approximation. The analysis techniques used are robust to mild violations in non-normality. Non-parametric methods were used in subsequent analyses. Changes during the GCBT condition were first analyzed groupwise to control for between-group differences, and then by combining the data of both groups (GCBT group and TAU + GCBT group; combined group). Further, changes in symptom ratings during follow-up and between the GCBT and TAU conditions were examined. Differences in CBCL and TRF symptom scores between time points were analyzed by Wilcoxon tests, and relative changes in scores between groups by Mann–Whitney U or Kruskal–Wallis -tests, depending on the grouping variable. In all testing, *p* values < 0.05 were considered statistically significant.

## 3. Results

### 3.1. Demographic Data

There were no differences in demographic data or baseline outcomes between the groups (Table 1) except for mothers´ education levels. Those in the GCBT group were more likely to have a university-educated mother than those in the TAU + GCBT group. Of the participants, 86.5% in the GCBT group and 82.3% in the TAU + GCBT group were known to follow the national core curriculum for education (data missing for other participants). The median age of all participants (*n* = 103) at baseline was 9.6 years (IQR = 8.7–11.1) and the most common diagnoses were ADHD (58.5%), depression or anxiety (37.2%), and other disorders of psychological development, mainly difficulties with reciprocal social interaction (28.7%). A total of 74.5% had more than one diagnosis.

All participants, for whom parent and/or teacher ratings were obtained at post-GCBT and/or at follow-up (treatment completers *n* = 88), attended seven or more GCBT sessions. There were no between-group differences in the number of attended children’s sessions categorized as 7–9 (13 children in the GCBT group, 10 in the TAU + GCBT group) or 10–12 sessions (32 children in the GCBT group, 33 in the TAU + GCBT group). However, there was a between-group difference (*p* = 0.043) in the number of attended parent sessions, categorized as zero (18 parents in the GCBT group, seven parents in the TAU + GCBT group), one (12 parents in the GCBT group, 14 in the TAU + GCBT group) or two (15 parents in the GCBT group, 22 in the TAU + GCBT group). Groups differed concerning 0 (*p* = 0.013) but not one or two attended sessions. Of the parents attending zero sessions, 15 in the GCBT group and one in the TAU + GCBT group participated in groups lacking parent sessions.

Of the participants for whom parent and/or teacher ratings were obtained at follow-up (*n* = 73), 45.2% were discharged from outpatient clinic services during the follow-up condition. There were no between-group differences in discharge rates during the follow-up condition.

### 3.2. Change in Parent- and Teacher-Rated Scores over Time

In linear mixed models, all participants showed reductions in CBCL-Int, CBCL-Ext, and CBCL-Tot from baseline to Time 2 (*p* < 0.001), from baseline to Time 3 (*p* < 0.001), and from baseline to Time 4 (*p* < 0.01), standardized betas ranging from −0.26 to −0.62 (Table 3). The model’s total explanatory power for predicting CBCL outcomes as a function of time was substantial (conditional R^2^s range from 0.68 to 0.79). However, participants showed no reductions in TRF-Int, TRF-Ext, or TRF-Tot from baseline to the subsequent time points.

### 3.3. Change in Parent- and Teacher-Rated Scores between Time Points and Groups

#### 3.3.1. The Effectiveness of GCBT

There were pre- to post-GCBT decreases in the CBCL-Ints, CBCL-Exts, and CBCL-Tots of the GCBT group, the TAU + GCBT group, and the combined group (Table 4). The CBCL-Int moved from the clinical to the normal range for the GCBT group (Mdn change 6.0) and from the borderline to the normal range for the TAU + GCBT group (Mdn change 3.0) and combined group (Mdn change 4.0). The CBCL-Ext moved from the clinical to the borderline range for the GCBT group (Mdn change 3.0) and combined group (Mdn change 3.5), and from the borderline to the normal range for the TAU + GCBT group (Mdn change 4.0). The CBCL-Tot moved from the clinical to the normal range for the combined group (Mdn change 5.0), from the clinical to borderline for the GCBT group (Mdn change 5.0), and from the borderline to the normal range for the TAU + GCBT group (Mdn change 5.0). There were no between-group differences (GCBT vs. TAU + GCBT group) in CBCL internalizing, externalizing, or total problem relative change scores during GCBT (subtraction of pre-GCBT scores from post-GCBT scores divided by pre-GCBT scores). The number of parent sessions attended was not associated with relative change in CBCL scores.

There were pre- to post-GCBT decreases in the TRF-Ints of the TAU + GCBT group and the combined group, the TRF-Ext of the GCBT group, and the TRF-Tots of the GCBT group and combined group (Table 4). The TRF-Int moved from the clinical to the borderline range for the TAU + GCBT (Mdn change 1.0) and the combined group (Mdn change 1.0). The TRF-Ext decreased but remained within the clinical range for the GCBT group (Mdn change 2.0), as did the TRF-Tot in both the GCBT group (Mdn change 1.0) and the combined group (Mdn change 2.0). There were no between-group differences in TRF internalizing, externalizing, or total problem relative change scores during GCBT.

#### 3.3.2. Maintenance of Effects during Follow-Up

Due to the lack of between-group differences in CBCL or TRF relative change scores during GCBT, we used the combined group for the follow-up comparisons. There were decreases in the CBCL-Int, CBCL-Ext, and CBCL-Tot from pre-GCBT to follow-up (Table 4). The CBCL-Int decreased but remained within the borderline range (Mdn change 1.5), and the CBCL-Ext and CBCL-Tot moved from the clinical to the borderline range (Mdn change 3.5 and 3.0, respectively). There was an increase in the CBCL-Int from post-GCBT to follow-up with symptoms, unfortunately, moving from the normal range to the borderline range (Mdn change 2.5), but no changes in the CBCL-Ext or CBCL-Tot. The number of parent-attended sessions was not associated with relative change in CBCL scores from pre-GCBT or post-GCBT to follow-up, except for a relative change from post-GCBT to follow-up in the CBCL-Tot, which was slightly higher in groups with zero parent-attended sessions compared to one parent-attended session. There were no decreases in TRF scores from pre-GCBT or post-GCBT to follow-up.

#### 3.3.3. Comparison of GCBT and the TAU Condition

There were decreases in the CBCL-Int, CBCL-Ext, and CBCL-Tot during the TAU condition (from pre- to post-TAU) of the TAU + GCBT group (Table 4) with symptom scores moving from the clinical to the borderline range (Mdn change 3.0, 6.0, and 5.0, respectively). There were no decreases in TRF scores.

There were no between-group differences (GCBT group during the GCBT condition vs. TAU + GCBT group during the TAU condition) in CBCL internalizing, externalizing, or total problem relative change scores. No between-group differences in relative change scores were found for the eight empirically based syndrome scales, either. Mothers´ education level, which differed between groups, was not associated with between-group outcomes with one exception. There was a difference in somatic complaints regarding participants with low-level maternal education, with symptom scores decreasing for the GCBT group during GCBT and increasing for the TAU + GCBT group during the TAU condition. Overall, the results are in favor of the combining of group data in the linear mixed model baseline analyses.

There were no between-group differences (GCBT group during GCBT vs. TAU + GCBT group during the TAU condition) in TRF internalizing, externalizing, or total problem relative change scores. No between-group differences in relative change scores were found for the eight empirically based syndrome scales, either.

Figure 2 depicts the pattern of change for internalizing and externalizing scores between time points.

## 4. Discussion

To our knowledge, this is the first study to examine the immediate and long-term effectiveness of GCBT delivered as part of routine services at specialized outpatient clinics in reducing parent- and teacher-rated internalizing and externalizing symptoms in children treated for mixed psychiatric disorders. Further, our study offers new insights into the potential utility of cost-effective, manualized GCBT by comparing its effects to individually tailored TAU in diverse pediatric, psychiatric populations.

Firstly, our results offer novel evidence for immediate positive outcomes of GCBT in the treatment of children with mixed psychiatric disorders exhibiting both internalizing and externalizing symptoms. Comprehensive symptom relief was demonstrated by the alleviation of parent-rated total problem symptoms from the clinical to the normal range (combined group) offering support for clinically significant change. Participants showed reductions in parent-rated internalizing and externalizing symptoms during GCBT. Symptom levels were reduced to a lower severity range in all group combinations (GCBT, TAU + GCBT, and combined group) from the borderline/clinical range to the normal range for internalizing symptoms and from the borderline/clinical range to the normal/borderline range for externalizing symptoms. Our findings are in line with prevailing research on CBT effects on internalizing [2,3,5,7] and externalizing symptoms, e.g., [16,17]. While positive effects of the use of anxiety-focused CBT in the treatment of comorbid externalizing symptoms in children with a primary diagnosis of anxiety have been demonstrated [24,25], our results also support its use in transdiagnostic populations of internalizing and externalizing disorders.

Our results suggested decreases also in teacher-rated internalizing symptoms from the clinical to the borderline range (combined group), offering to the best of our knowledge new data on the effectiveness of GCBT conducted in treatment settings using teachers as informants. Our findings were not in line with earlier findings [13,16] of improvements in teacher-rated externalizing symptoms (combined group). However, despite the differences in magnitude, teacher-reported symptoms followed a similar trajectory as parent-reported symptoms. Further, previous work has indicated that teachers report less internalizing and externalizing symptoms in children than their caregivers [50,51]. Results of teacher reports must in all be interpreted with caution as over 30% of the participants in both groups experienced at least one change in teachers during the study which may have added to response variability between time points. Although we could not measure inter-teacher reliability because the TRF questionnaires were completed at four different time points, correlations of ratings between pairs of teachers tend to be large [52]. While changes in informants pose problems for the interpretation of results, it can be hard to avoid in a study that spans over grade levels.

Secondly, participants showed substantial reductions in all parent-rated symptoms over time from entry to the study to subsequent time points. Our findings suggested long-term GCBT outcomes particularly for externalizing symptoms. Symptom severity was lower at the 6-month follow-up than at pre-GCBT, and post-GCBT gains were maintained at the 6-month follow-up (combined group). Our results support prior data [15,17,18] regarding long-term CBT effects in the treatment of children with behavioral problems. Parent-rated internalizing symptoms also decreased from pre-GCBT to the 6-month follow-up offering partial support for long-term effects. However, gains were not maintained from post-GCBT to the 6-month follow-up, a result differing from prior findings [3]. In previous studies conducted on FRIENDS in a treatment setting, gains have either been maintained or increased at the 12-month follow-up [37,38,39]. However, due to the shorter follow-up condition in our study, it is not possible to reliably predict how symptoms could potentially have varied or stabilized at the 12-month follow-up. Further, in past studies booster sessions were conducted post-GCBT and not during the GCBT condition as in our study, suggesting guided rehearsal of CBT techniques also during a follow-up. Hence, it could be interpreted that the implementation of periodic follow-up sessions after the end of GCBT may help in maintaining treatment outcomes and be recommendable. No changes in teacher-rated symptom levels were found in any follow-up comparisons.

While the number of parent sessions attended differed between groups, it was not associated with relative change in parent-rated internalizing or externalizing symptom levels during GCBT or follow-up. The results are in line with previous findings of parental involvement not moderating post-CBT outcomes in the treatment of internalizing disorders [53], although it has been suggested to be associated with better long-term outcomes [53,54]. Parenting programs, on the other hand, are established and recommended psychosocial interventions for antisocial behavior and conduct disorders in children [55]. Due to the small number of parent sessions and limited data of treatment completers in our sample, strong conclusions related to parental participation cannot be drawn based on our results. Parental involvement (parent sessions, taking part in homework, or use of workbook), as opposed to no teacher involvement in the intervention, may partly be related to the discrepancy between parent and teacher reports. It may have helped parents to be more sensitive to any changes in their children’s symptoms.

Thirdly, our results suggested no differences in effectiveness between the GCBT and the TAU condition in parent- or teacher-rated symptoms. As with GCBT, parents reported internalizing, externalizing, and total symptom alleviation also during the TAU condition. Symptom levels shifted from the clinical to the borderline range. While the GCBT intervention offered systematic, structured sessions and broader use of predefined therapeutic techniques in group form, the TAU services provided more individually tailored but generally less frequent treatment sessions for each child, considering the specific needs of each child. The lack of difference in the reduction of symptoms may be partly due to both conditions providing quality psychoeducation. The results support previous findings of no clear evidence of CBT being more effective than TAU in the treatment of internalizing disorders [3,31]. However, the TAU services provided in our study cannot be directly compared to the content and format of TAU conditions in previous studies on internalizing or externalizing symptoms. The results should also be interpreted with caution due to possible placebo effects. Parents may have experienced anticipation and relief regarding their children’s referral to GCBT. This may have contributed to reductions in the symptoms they observed during the TAU condition, as parents’ psychological symptoms have been found to affect their ratings of children’s internalizing and externalizing behavior [56]. Further, the baseline assessment may have increased parents’ understanding of their children’s symptoms, affecting their parenting practices and allocation of attention during the TAU condition, which in turn may have reduced their children’s symptoms. Although teacher-rated internalizing symptoms were alleviated during GCBT but not during the TAU condition, no between-group differences were observed for the two conditions. This may be due in part to the small sample sizes in the group comparisons.

Findings of similar effects of structured, manualized GCBT and individually tailored TAU in the treatment of children with mixed psychiatric disorders exhibiting borderline to clinical range internalizing and externalizing symptoms provide support for the increased implementation of GCBT, with careful consideration on a case-by-case basis. Implementation of GCBT in settings other than specialized child psychiatric clinics, to which access is very limited, is also supported. This view is congruent with a recent study [57] in which the FRIENDS program showed corresponding outcomes in reducing anxiety, depression, and conduct problems when implemented by less specialized personnel at schools or by trained mental health professionals in children´s mental health community clinics [57]. The FRIENDS program has been shown to be an effective school-based anxiety prevention program [58], which is also endorsed by the World Health Organization [59]. Our findings on the immediate effects of the program on internalizing symptoms, as rated by parents and teachers, and on the immediate and long-term effects on externalizing symptoms, as rated by parents, support the increasing use of GCBT at primary health care levels for children with a variety of problem behaviors. This could enable a better allocation of the scarce resources of specialized mental health care. Expanding primary care services has been identified as a key means of improving access to mental health services and closing the treatment gap [60]. As manualized GCBT requires relatively short training, fewer staff resources per child, and is typically less expensive than individual treatments, its more widespread implementation is justified. Our findings are clinically relevant given the increasing occurrence of mental health problems in children [33] and the global shortage of human and financial resources allocated to mental health care [32].

Several limitations of the present study are noteworthy. First, there was a significant increase in missing data towards the end of the study, and as a result, some statistical power was lost especially in teacher ratings of symptoms. Further, the results should be interpreted with caution due to the testing of multiple scores. We did not adjust the statistical significance of our results for the number of tests (Bonferroni method), because our study was a hypothesis testing study. As suggested by Perneger [61], the Bonferroni method applies to the general null hypothesis (that all null hypotheses are true simultaneously), which is seldom of interest or use to researchers. Perneger [61] also stated that describing what significance tests have been performed, and why, is usually the best way to deal with multiple comparisons. Second, the study would have been improved by measures of treatment fidelity and adherence. Third, possible dependencies among observations within the 31 different GCBT groups delivered were not accounted for. In addition, we are aware of the potential bias due to non-random grouping. Due to the practical constraints of treatment implementation in a naturalistic clinical setting and limited sample size, the above caveats were overlooked and their impact on the results is unknown. Fourth, although there were no between-group differences in terms of medication changes, the contribution of these changes to treatment outcomes was not assessed. Further, possible developmental factors related to different psychiatric diagnoses and symptoms, and their potential association with treatment outcomes may influence the generalizability of our results. Younger age in children has been associated with larger post-CBT externalizing symptom reductions [17], and externalizing problem behaviors have been suggested to become more resistant to change with age [62,63]. Finally, as treatment took place in a care setting, a no-treatment waitlist control group was not used, which may have provided a more accurate reference for the effectiveness of GCBT.

Nevertheless, the present study shows high external validity as both GCBT and TAU were delivered in outpatient clinics following each child´s treatment plan. Further, our study adds to the limited research on GCBT effectiveness in the treatment of children with psychiatric comorbidities as more than 70% of our participants had multiple psychiatric disorders. In the modern conceptualization of psychopathology as a dimensional framework [29], signs and symptoms of disorders in diagnostic categories can be grouped into internalizing and externalizing dimensions. Dimensional grouping is sensitive in tracking changes, especially in the co-occurrence of symptoms of different categorical diagnoses [29]. Future work could examine possible interactions between alleviations in internalizing and externalizing symptoms. It would also be of interest to assess which aspects of treatment are relevant for children with multidimensional psychiatric symptoms, for example, which specific CBT strategies and techniques contribute to symptom relief.

## 5. Conclusions

Our results indicated that GCBT delivered in naturalistic, clinical settings was effective in reducing internalizing and externalizing symptoms in children treated for mixed psychiatric disorders. Parent ratings demonstrated significant GCBT benefits in both groupings of disorders, and teacher ratings suggested novel findings of GCBT effects in the treatment of internalizing symptoms. Parent-rated long-term gains were prominent especially for externalizing symptoms. Moreover, our findings suggested equal effectiveness of manualized GCBT and individually tailored TAU services. This in turn provides support for the wider use of cost-effective, manualized GCBT at primary health care levels in the treatment of children with borderline to clinical range psychiatric symptoms, to better meet the increased mental health needs of children and the limited resources of mental health services.

## Figures and Tables

**Figure 1 children-09-01602-f001:**
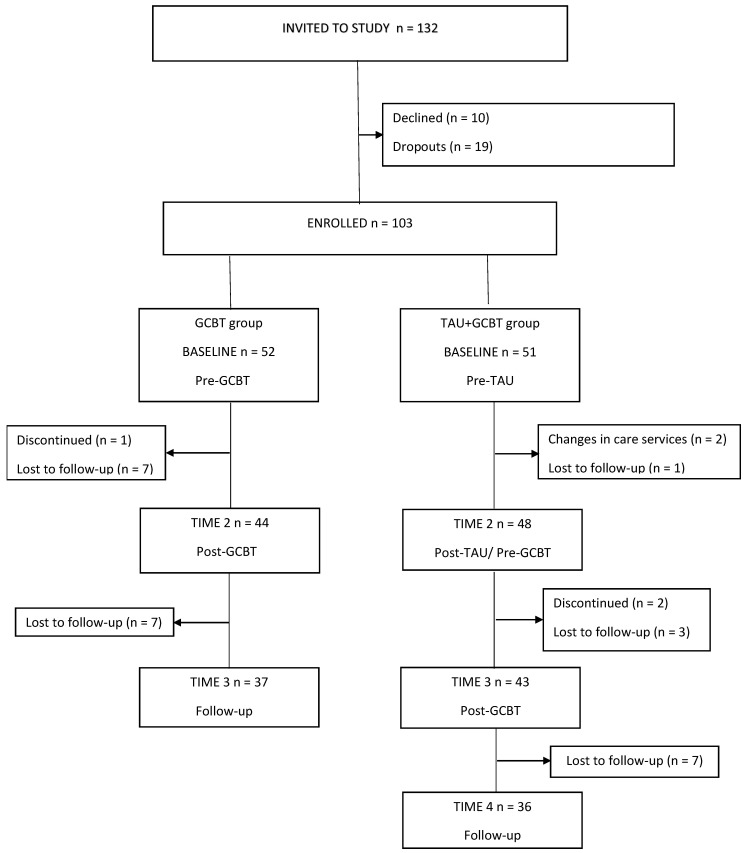
Study flowchart. Legend: From 132 subjects invited to the study 103 were enrolled. At each time point, parent- and teacher-rated questionnaires were completed in both study groups.

**Figure 2 children-09-01602-f002:**
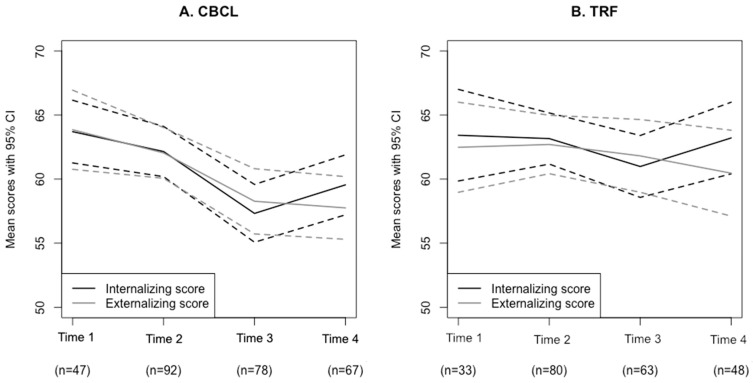
Change in (**A**) CBCL and (**B**) TRF outcomes between time points. Note. Time 1 = Pre-TAU (TAU + GCBT group); Time 2 = Pre-GCBT, Time 3 = Post-GCBT, and Time 4 = Follow-up (Combined group).

**Table 1 children-09-01602-t001:** Demographic data for the GCBT (*n* = 52) and TAU + GCBT group (*n* = 51) at baseline.

	GCBT Group	TAU + GCBT Group
Gender		
Boys *n* (%)	38 (73.1)	32 (62.7)
Girls *n* (%)	14 (26.9)	19 (37.3)
Age in years, median (Q_1_–Q_3_)	9.4 (8.5–10.9)	10.0 (8.8–11.5)
Multiple psychiatric comorbidity ^a^ *n* (%)	35 (72.9)	35 (76.1)
ADHD ^a^ *n* (%)	29 (60.4)	26 (56.5)
Depression or anxiety ^a^ *n* (%)	18 (37.5)	17 (37.0)
Other disorders of psychological development ^ab^ *n* (%)	12 (25.0)	15 (32.6)
Conduct disorder ^a^ *n* (%)	12 (25.0)	5 (10.9)
Autism spectrum disorder ^a^ *n* (%)	9 (18.8)	7 (15.2)
Specific developmental disorders ^ac^ *n* (%)	9 (18.8)	6 (13.0)
Tic disorder ^a^ *n* (%)	3 (6.3)	5 (10.9)
Mother’s education level ^d^ *n* (%)		
Low level	12 (23.5)	15 (31.3)
Mid-level	18 (35.3)	25 (52.1)
High level	21 (41.2)	8 (16.7)
Family type ^a^ *n* (%)		
Nuclear family	26 (54.2)	22 (47.8)
Single-parent family	21 (43.8)	20 (43.5)
Adoption/foster care	1 (2.1)	4 (8.7)
Medication change ^a^ *n* (%)	14 (29.2)	13 (28.3)
Change in teachers ^e^ *n* (%)	16 (31.4)	21 (44.7)
Child welfare notification/service ^a^ *n* (%)	1 (2.1)	5(10.9)
CBCL-Int, M (SD)	63.8 (9.8)	63.7 (8.3)
CBCL-Ext, M (SD)	64.1 (8.3)	63.9 (10.5)
CBCL-Tot, M (SD)	65.7 (7.2)	65.7 (8.2)
TRF-Int, M (SD)	63.1 (9.5)	63.4 (10.1)
TRF-Ext, M (SD)	63.8 (10.7)	62.5 (9.9)
TRF-Tot, M (SD)	67.1 (9.5)	66.1 (9.1)

Note: ^a^ Missing data for 8.7% (7.7% in the GCBT group, 9.8% in the TAU + GCBT group). ^b^ Mainly difficulties with reciprocal social interaction. ^c^ Diagnoses F80, F81 or F83 (ICD-10. 2016). ^d^ Education level: low level = comprehensive or secondary education, mid-level = upper vocational education, high level = university education. Missing data for 3.9% (1.9 in the GCBT group, 5.9 in the TAU + GCBT group). ^e^ Missing data for 4.9% (1.9 in the GCBT group, 7.8 in the TAU + GCBT group). Abbreviations: Q_1_–Q_3_, inter-quartile ranges; ADHD, attention deficit hyperactivity disorder; CBCL-Int, CBCL internalizing score; CBCL-Ext, CBCL externalizing score; CBCL-Tot, CBCL Total problem score; TRF-Int, TRF Internalizing score; TRF-Ext, TRF externalizing score; TRF-Tot, TRF total problem score; M, mean; SD, standard deviation.

**Table 2 children-09-01602-t002:** Missing CBCL and TRF scores at the different time points.

	GCBT Group (*n* = 52)						TAU + GCBT Group (*n* = 51)							
	CBCL-Scores			TRF-Scores			CBCL-Scores				TRF-Scores			
	T1	T2	T3	T1	T2	T3	T1	T2	T3	T4	T1	T2	T3	T4
Present	51	41	32	43	36	28	47	41	37	35	33	37	27	20
Missing (%)	1 (1.9)	11 (21.2)	20 (38.5)	9 (17.3)	16 (30.8)	24 (46.2)	4 (7.8)	10 (19.6)	14 (19.6)	16 (31.4)	18 (35.3)	14 (27.5)	24 (47.1)	31 (60.8)

Note: T1 = Time 1 (pre-GCBT for the GCBT group and pre-TAU for the TAU + GCBT group), T2 = Time 2 (post-GCBT for the GCBT group and post-TAU for the TAU + GCBT group), T3 = Time 3 (6-month follow-up for the GCBT group and post-GCBT for the TAU + GCBT group), T4 = Time 4 (6-month follow-up for the TAU + GCBT group).

**Table 3 children-09-01602-t003:** CBCL and TRF score changes over time from baseline to Time 2, Time 3, and to Time 4.

	CBCL Int		CBCL Ext		CBCL Tot		TRF Int		TRF Ext		TRF Tot	
Predictor	Estimates (95% CI)	*p*	Estimates (95% CI)	*p*	Estimates (95% CI)	*p*	Estimates (95% CI)	*p*	Estimates (95% CI)	*p*	Estimates (95% CI)	*p*
Intercept (Baseline)	63.71 (61.85–65.58)	<0.001	63.65 (61.71–65.58)	<0.001	65.56 (63.96–67.17)	<0.001	63.68 (61.64–65.73)	<0.001	63.16 (60.98–65.34)	<0.001	66.68 (64.79–68.57)	<0.001
Time 2	−4.13 (−5.81–−2.46)	<0.001	−2.74 (−4.18–−1.31)	<0.001	−3.52 (−4.90–−2.15)	<0.001	−0.97 (−3.05–1.12)	0.365	−1.04 (−2.91–0.83)	0.277	−1.64 (−3.40–0.12)	0.068
Time 3	−4.75 (−6.54–−2.96)	<0.001	−4.88 (−6.42–−3.34)	<0.001	−5.23 (−6.71–−3.76)	<0.001	−1.02 (−3.30–1.25)	0.378	−0.68 (−2.72–1.35)	0.512	−1.06 (−2.98–0.86)	0.281
Time 4	−3.68 (−6.01–−1.36)	0.002	−6.54 (−8.54–−4.53)	<0.001	−5.38 (−7.30–−3.47)	<0.001	−0.66 (−4.07–2.76)	0.706	−2.70 (−5.77–0.37)	0.085	−2.48 (−5.37–0.41)	0.092
Random Effects												
σ^2^	30.41		22.13		20.49		37.56		29.32		26.35	
τ_00_	60.42 _id_		77.02 _id_		47.18 _id_		52.65 _id_		77.39 _id_		52.25 _id_	
ICC	0.67		0.78		0.70		0.58		0.73		0.66	
N	103 _id_		103 _id_		103 _id_		96 _id_		96 _id_		96 _id_	
Observations	284		284		284		224		224		224	
Marginal R^2^/Conditional R^2^	0.045/0.680		0.053/0.789		0.070/0.718		0.002/0.585		0.005/0.727		0.008/0.668	

Note. Abbreviations: CBCL-Int, CBCL internalizing score; CBCL-Ext, CBCL externalizing score; CBCL-Tot, CBCL Total problem score; TRF-Int, TRF Internalizing score; TRF-Ext, TRF externalizing score; TRF-Tot, TRF total problem score; M, mean; SD, standard deviation; σ^2^, variance of model; τ_00_, variance between individuals; ICC, intraclass correlation; N, number; _id_, cluster.

**Table 4 children-09-01602-t004:** Descriptive statistics of the outcome measures at different time points.

Group	Time Point	Measure Mean (SD) Median (IQR)					
		CBCL-Int	CBCL-Ext	CBCL-Tot	TRF-Int	TRF-Ext	TRF-Tot
GCBT	Pre-GCBT	63.8 (9.8) 65.0 (58.0–71.0)	64.1(8.3) 64.0 (60.0–70.0)	65.7 (7.2) 66.0 (61.0–72.0)	63.1 (9.5) 64.0 (57.0–69.0)	63.8 (10.7) 66.0 (55.0–73.0)	67.1 (9.5) 67.0 (60.0–75.0)
	Post-GCBT	57.3 (10.3) 59.0 (51.0–65.0)	61.1 (9.6) 61.0 (53.5–67.0)	60.7 (8.6) 61.0 (53.0–65.5)	61.5 (10.1) 61.5 (56.0–69.0)	62.7 (12.0) 64.0 (54.3–71.0)	64.6 (10.4) 66.0 (60.3–71.5)
	Follow-up	60.0 (9.9) 63.5 (52.5–67.8)	60.7 (8.9) 62.0 (51.3–67.8)	61.0 (8.6) 63.0 (54.5–67.5)	63.4 (9.1) 64.5 (58.0–68.8)	63.1 (12.3) 62 (53.5–74.5)	66.5 (9.4) 64.0 (59.0–75.0)
TAU + GCBT	Pre-TAU	63.7 (8.3) 64.0 (58.0–69.0)	63.9 (10.5) 66.0 (54.0–72.0)	65.7 (8.2) 68.0 (61.0–71.0)	63.4 (10.1) 64.0 (57.5–70.0)	62.5 (9.9) 63.0 (58.5–69.0)	66.1 (9.1) 67.0 (58.5–73.0)
	Post-TAU/ Pre-GCBT	60.1 (8.7) 61.0 (53.0–66.5)	59.5 (10.4) 60.0 (50.5–68.5)	61.7 (8.1) 63.0 (56.0–68.0)	63.2 (8.4) 64.0 (58.0–68.0)	61.5 (9.6) 63.0 (57.0–69.0)	64.9 (7.5) 67.0 (58.5–70.0)
	Post-GCBT	57.4 (9.8) 58.0 (49.0–63.5)	55.1 (12.3) 56.0 (44.0–63.5)	58.1 (9.2) 58.0 (51.5–61.5)	60.3 (9.0) 63.0 (56.0–67.0)	60.7 (10.4) 61.0 (53.0–69.0)	62.7 (8.7) 63.0 (54.0–70.0)
	Follow-up	59.2 (9.5) 59.0 (52.0–66.0)	55.1 (10.4) 56.0 (48.0–62.0)	59.1 (7.6) 60.0 (53.0–65.0)	63.0 (10.6) 62.5 (59.0–72.5)	56.8 (9.6) 57.0 (49.0–67.0)	61.6 (8.6) 64.5 (57.0–67.0)
Combined	Pre-GCBT	62.15 (9.5) 62.5 (54.0–69.8)	62.0 (9.6) 63.5 (56.5–70.0)	63.9 (7.8) 64.0 (58.3–71.0)	63.2 (9.0) 64.0 (58.0–68.8)	62.7 (10.2) 64.5 (57.0–71.0)	66.1 (8.6) 67.0 (59.3–72.0)
	Post-GCBT	57.3 (10.0) 58.5 (50.0–64.3)	58.3 (11.3) 60.0 (51.0–65.0)	59.4 (8.9) 59.0 (53.0–65.0)	61.0 (9.6) 63.0 (56.0–68.0)	61.8 (11.3) 63.0 (53.0–70.0)	63.8 (9.7) 65.0 (57.0–70.0)
	Follow-up	59.6 (9.6) 61 (52.0–67.0)	57.8 (10.0) 60.0 (51.0–65.0)	60.0 (8.1) 61 (54.0–66.0)	63.2 (9.6) 63.5 (58.3–69.8)	60.5 (11.5) 59.5 (53.0–69.8)	64.5 (9.3) 64.5 (59.0–70.8)

Note. Abbreviations: SD, standard deviation; IQR, inter-quartile range; CBCL-Int, CBCL internalizing score; CBCL-Ext, CBCL externalizing score; CBCL-Tot, CBCL total problem score; TRF-Int, TRF internalizing score; TRF-Ext, TRF externalizing score; TRF-Tot, TRF total problem score.

## Data Availability

Sarianna Barron-Linnankoski takes responsibility for the integrity of the data and the accuracy of the data analysis.

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
