# Peer review of "Effectiveness of Group CBT on Internalizing and Externalizing Symptoms in Children with Mixed Psychiatric Disorders"

_children, 2022, doi:10.3390/children9111602_

Round 1
Reviewer 1 Report
The paper entitled "Effectiveness of group CBT on internalizing and externalizing 2 symptoms in children with mixed psychiatric disorders" is an interesting work that tested the efficacy of group cognitive behavioral therapy in children with mixed psychiatric disorders and presented internalizing and externalizing symptoms. As stated by the authors, the topic is relevant because no study tested the effectiveness of group CBT in a naturalistic context in children with mixed psychiatric disorders with internalizing and externalizing problems. Nevertheless, this study presents several issues. Particularly, I would suggest to the authors an extensive revision of the introduction section, explaining the rationale and deepening the theoretical background, leading the reader to understand better state of the art and the motivation of their work. Furthermore, the method is not clearly explained and needs some clarifications. Please find below my specific comments:
Abstract:
A brief paragraph on the theoretical background is missing in the first part of the abstract. Furthermore, when an abbreviation is presented, it should be explained; it is not clear at first reading what GCBT is. Finally, in the conclusion authors stated: "providing support for the wider use of 31 cost-effective manualized GCBT also at lower levels of mental health care in the treatment of children with a broad range of psychiatric symptoms.". The logic connection with the results is unclear; I would suggest rephrasing and clarifying the logic steps that lead to the conclusion.
Introduction:
The authors stated: "cognitive-behavioral therapy (CBT) is shown to be effective in the treatment of children and adolescents (children) with internalizing and externalizing disorders. Although 39 comorbidities are common within and between internalizing and externalizing groupings 40 [1-4], there is a gap in the research on CBT in the treatment of children with a wider range 41 of psychiatric disorders exhibiting both internalizing and externalizing symptoms." I would suggest rephrasing this sentence because there are a lot of repetitions in it, making it difficult to read the paragraph fluently. Furthermore, the first sentence needs a reference.
From lines 42-48: In these lines, the authors introduce the aims of their work but a deepening of the aims is missing. I think that before introducing the aims of the research, authors should describe their rationale, and at the end of the paragraph, they can state their aims and hypotheses clearer.
Furthermore, I would suggest to the authors to describe the main strategies of CBT to which they refer. This could help the reader better understand the topic and the work's rationale. Additionally, a more detailed clinical description of the psychiatric children with internalizing and externalizing symptoms that are this paper's subject of interest may be appropriate to understand the rationale of the intervention.
Method:
In the participants' paragraph, it is not clear if the participants were or were not randomized to the two groups, and the method of randomization should be explained.
Furthermore, eligibility and exclusion criteria are stated, but how they were assessed and applied is unclear.
Figure 1 should be revised and ameliorated. (e.g., the statement "Informed written consent not returned" it is unclear what it refers to.
The timeline and the procedure for the two groups is not clear.
I think avoiding vague sentences (e.g., "multiple diagnoses") is better. This makes not clear what the authors are talking about.
The authors described the sample, but not all instruments were described; they should be added in the measures paragraph.
I would suggest adding a table or a description for a detailed description of the two treatments (or adding an explanation in the introduction section).
Results and discussion:
In the tables, legends are missing.
Authors stated: "Results of teacher reports must in all be interpreted with caution as over 30 % of the 398 participants in both groups experienced at least one change in teachers during the study 399 which may have added to response variability between time points. While this poses 400 problems for the interpretation of results, it can be hard to avoid in a study that spans over 401 grade levels." This is not clear and could confuse the reader. Please clarify in the method.
The interpretation of the results is still not clear. Adding a detailed description of the interventions, their strategies, and the rationale could help the reader to follow the discussion of the results better.
I would suggest that the authors add the clinical implications of their results and future research suggestions.
This sentence is unclear: "Our results indicated effectiveness of GCBT in naturalistic, clinical settings in the 495 treatment of children with mixed psychiatric disorders exhibiting internalizing and exter- 496 nalizing symptoms." Effectiveness compared to what? Which aspects are functional for this kind of patients? Which components worked and which did not?
Author Response
Dear Editor and Reviewers: Thank you for your interest and for your comments concerning our manuscript entitled “Effectiveness of group CBT on internalizing and externalizing symptoms in children with mixed psychiatric disorders”. Your valuable comments have been of great help to us in revising and improving our paper. We have studied your comments carefully and have made the requested corrections and additions which we hope will meet with your approval. We have also had the text proofread for English language and style. Please kindly view the changes in our uploaded revised manuscript.
The main modifications in the paper and the responses to the reviewers´ comments are as follows:
Response to Reviewer 1:
Comment 1:
Abstract:
A brief paragraph on the theoretical background is missing in the first part of the abstract. Furthermore, when an abbreviation is presented, it should be explained; it is not clear at first reading what GCBT is. Finally, in the conclusion authors stated: "providing support for the wider use of 31 cost-effective manualized GCBT also at lower levels of mental health care in the treatment of children with a broad range of psychiatric symptoms.". The logic connection with the results is unclear; I would suggest rephrasing and clarifying the logic steps that lead to the conclusion.
Response to comment 1:
Thank you for your comment. We have now clarified the abbreviation of GCBT and added a brief introduction to the theoretical background regarding the gap in research in this sample population. We have also tried to clarify the logical steps in the Conclusions section of the abstract.
Introduction section
Comment 2:
The authors stated: "cognitive-behavioral therapy (CBT) is shown to be effective in the treatment of children and adolescents (children) with internalizing and externalizing disorders. Although 39 comorbidities are common within and between internalizing and externalizing groupings 40 [1-4], there is a gap in the research on CBT in the treatment of children with a wider range 41 of psychiatric disorders exhibiting both internalizing and externalizing symptoms." I would suggest rephrasing this sentence because there are a lot of repetitions in it, making it difficult to read the paragraph fluently. Furthermore, the first sentence needs a reference.
From lines 42-48: In these lines, the authors introduce the aims of their work but a deepening of the aims is missing. I think that before introducing the aims of the research, authors should describe their rationale, and at the end of the paragraph, they can state their aims and hypotheses clearer.
Response to comment 2: We realized from your and the other reviewer's valuable comments that the first paragraph of the original introduction was not clearly outlined. The paragraph itself contained repetitions. Moreover, the same things are demonstrated with more depth in the following paragraphs. Based on your constructive comments, we have removed the first paragraph in its entirety to help the readability of the introduction section. We believe that this will enable us to provide the reader with a more coherent understanding of the background, rationale, and aims of the study.
Comment 3: Furthermore, I would suggest to the authors to describe the main strategies of CBT to which they refer. This could help the reader better understand the topic and the work's rationale.
Response to comment 3: Thank you for your advice to describe the main strategies of CBT. We have added the following sentence to the last paragraph of the introduction:
The FRIENDS program follows the main principles of CBT, involving techniques and approaches that build an understanding of the link between our thoughts, feelings, and behavior. It focuses on reducing current symptomology and on promoting resilience and well-being.
Further, in the 2.4 Treatment section, we have added to the description of CBT-specific techniques used in the Friends program as follows:
FRIENDS is an acronym symbolizing and helping children to remember the topics covered in the program: F = Feelings; R = Relaxation; I = Inner thoughts; E = Exploring strategies; N = Now reward yourself; D = Don´t forget to practice; S = Stay calm. The therapeutic techniques of FRIENDS include psychoeducation, empathy training and self-regulation, relaxation and mindfulness, cognitive strategies, exposure, social support training, self-rewards, and relapse prevention. These topics and techniques were addressed in a progressive and integrative way throughout the program.
Comment 4: Additionally, a more detailed clinical description of the psychiatric children with internalizing and externalizing symptoms that are this paper's subject of interest may be appropriate to understand the rationale of the intervention.
Response to comment 4:
Following your and the other reviewer's helpful comment, we have tried to clarify (in the third last and in the last paragraph of the Introduction section) that our population sample of interest are children with a range of psychiatric disorders, with symptoms on both the internalizing and externalizing dimensions.
We have also described the diagnostic processes used in more detail at the end of the 2.3 Measures section.
Method section
Comment 5:
In the participants' paragraph, it is not clear if the participants were or were not randomized to the two groups, and the method of randomization should be explained.
Response to comment 5:
Thank you for pointing out to us this unclearness related to whether the participants were randomized to the groups. We have now removed the word “Randomized” from the abstract and from Figure 1, which was incorrect. We have also slightly modified the sentence explaining the assignment to groups in the “Participant” paragraph as follows: “Assignment to the GCBT or TAU+GCBT group was determined by the order in which children were referred to GCBT and the order in which GCBT groups were initiated in the age range in question.” Further, we have moved the sentence to the beginning of the paragraph, where the assignment into groups was already mentioned. Assignment practices were related to the treatment process, not the study. Thus, we understand that participants were not randomized to groups. Finally, we have added the following italicized sentence to the Discussion section concerning the limitations of the study: “Third, possible dependencies among observations within the 31 different GCBT groups delivered were not accounted for. In addition, we are aware of the potential bias due to non-random grouping. Due to the practical constraints of treatment implementation in a naturalistic clinical setting and limited sample size, the above caveats were overlooked and their impact on the results is unknown. “
Comment 6:
Furthermore, eligibility and exclusion criteria are stated, but how they were assessed and applied is unclear.
Response to comment 6
Thank you for your comment. We have now tried to further define the eligibility and exclusion criteria in the 2.1 Participants section as follows:
“Exclusion criteria to enable skill training were children’s excessive physical aggression and excessive physical restlessness. Referral and exclusion criteria were based on clinical and diagnostic assessments carried out by physician-led multidisciplinary teams (e.g., thorough clinical evaluation of the child, extensive data collection, and the use of diagnostic, structured methods; see 2.3. Measures section). All children referred to GCBT in outpatient clinics were invited to the study. Children who were referred to GCBT but whose severity of psychiatric symptoms required inpatient treatment (e.g. acute suicidality) were not included in the study because the GCBT protocol was largely modified in inpatient units to better suit the treatment setting.”
We have further explained the diagnostic processes conducted in the 2.3 Measures section.
Comment 7:
Figure 1 should be revised and ameliorated. (e.g., the statement "Informed written consent not returned" it is unclear what it refers to.
Response to comment 7: Thank you for asking us to clarify Figure 1 regarding the statement “Informed written consent not returned”. We have now corrected Figure 1. The children who were referred to as “Excluded” were two children/families who failed to return written consent, although they filled all the questionnaires of the study. These two children were not included in the study. We have now added them to those children/ families who `Declined` from the study.
We have also tried to explain the issues related to written informed consent more clearly in the text (second paragraph in section 2.1 Participants).
Comment 8: The timeline and the procedure for the two groups is not clear.
Response to comment 8:
Thank you for asking us to clarify our setup regarding the TAU and GCBT treatments. We have now tried to clarify the timeline and procedure in the text (e.g., 2.2. Procedure paragraph, 2.4 Treatment paragraph), in Figure 1 and in the tables by defining in more detail what treatments are referred to at the different time points.
The TAU+GCBT group first received TAU followed by GCBT. The GCBT group received GCBT immediately. We have assessed symptom relief in both treatment conditions, TAU and GCBT. With regard to GCBT, we have assessed its effectiveness in both groups separately and also by combining the groups. Further, we compared the effectiveness of the TAU and GCBT conditions. We hope that our changes will help the reader to better understand the setup.
Comment 9: I think avoiding vague sentences (e.g., "multiple diagnoses") is better. This makes not clear what the authors are talking about.
Response to comment 9:
Thank you for your comment. We have now added a description of the conduction of the diagnostic processes at the end of the 2.3 Measures section. Further, we have rephrased the use of “multiple diagnoses” to psychiatric comorbidities.
Comment 10: The authors described the sample, but not all instruments were described; they should be added in the measures paragraph.
Response to comment 10:
Thank you for your observant comment regarding all instruments not described. We have now added the information at the end of the 2.3. Measures paragraph.
Comment 11:
I would suggest adding a table or a description for a detailed description of the two treatments (or adding an explanation in the introduction section).
Response to comment 11:
Thank you for your advice to describe the two treatments, TAU and GCBT, in more detail. We have now made the following changes:
Addition to the last paragraph of the Introduction section:
The FRIENDS program follows the main principles of CBT, involving techniques and approaches that build an understanding of the link between our thoughts, feelings, and behavior. It focuses on reducing current symptomology and on promoting resilience and well-being.
Additions to Section 2.4 Treatments (of the Methods):
First paragraph: description of TAU added/moved here.
Second paragraph: More detailed description of the FRIENDS program:
“FRIENDS is an acronym symbolizing and helping children to remember the topics covered in the program: F = Feelings; R = Relaxation; I = Inner thoughts; E = Exploring strategies; N = Now reward yourself; D = Don´t forget to practice; S = Stay calm. The therapeutic techniques of FRIENDS include psychoeducation, empathy training and self-regulation, relaxation and mindfulness, cognitive strategies, exposure, social support training, self-rewards, and relapse prevention. These topics and techniques were addressed in a progressive and integrative way throughout the program.”
Further, we have tried to clarify the timeline and treatments of the GCBT and TAU+GCBT groups in Figure 1, the tables, and section 2.2. Procedure paragraph.
We are happy to add a table describing content for each FRIENDS session, if required.
Results and Discussion sections:
Comment 12: In the tables, legends are missing.
Response to comment 12:
Thank you for your comment. We have now added the requested information at the Note sections of the tables.
Comment 13: Authors stated: "Results of teacher reports must in all be interpreted with caution as over 30 % of the 398 participants in both groups experienced at least one change in teachers during the study 399 which may have added to response variability between time points. While this poses 400 problems for the interpretation of results, it can be hard to avoid in a study that spans over 401 grade levels." This is not clear and could confuse the reader. Please clarify in the method.’
Response to comment 13:
Thank you for your comment. We have tried to clarify this issue by modifying the text as follows.
Addition to the end of 2.3. Measures section:
The information on teacher changes was obtained from the TRF.
Addition to Discussion section:
Results of teacher reports must in all be interpreted with caution as over 30 % of the participants in both groups experienced at least one change in teachers during the study which may have added to response variability between time points. Although we could not measure inter-teacher reliability because the TRF questionnaires were completed at four different time points, correlations of ratings between pairs of teachers tend to be large [50]. While changes in informants pose problems for the interpretation of results, it can be hard to avoid in a study that spans over grade levels.
Comment 14:
The interpretation of the results is still not clear. Adding a detailed description of the interventions, their strategies, and the rationale could help the reader to follow the discussion of the results better.
I would suggest that the authors add the clinical implications of their results and future research suggestions.
Response to comment 14:
Thank you for your constructive comment. We hope that the ameliorated description of the set-up, procedure, and treatments help the reader to follow the discussion of the results. We have now added the following sentence to the Discussion section in relation to the clinical implements of our results:
Our results also have implications for the global mental health staff shortage and cost-effectiveness. As GCBT requires fewer staff resources per child and is typically less expensive than individual treatments, its more widespread implementation is warranted.
Please see response to comment 15 with regard to future research suggestions.
Comment 15:
This sentence is unclear: "Our results indicated effectiveness of GCBT in naturalistic, clinical settings in the 495 treatment of children with mixed psychiatric disorders exhibiting internalizing and exter- 496 nalizing symptoms." Effectiveness compared to what? Which aspects are functional for this kind of patients? Which components worked and which did not?
Response to comment 15:
Thank-you for your comment.
We have rephrased the mentioned unclear sentence as follows:
“Our results indicated that GCBT delivered in naturalistic, clinical settings was effective in reducing internalizing and externalizing symptoms in children treated for mixed psychiatric disorders.”
- As children's symptom levels were reduced to a lower severity range in internalizing, externalizing and comprehensive symptoms we suggest our findings may offer support for clinically significant change.
As the topics, strategies, and techniques of the FRIENDS program are addressed in a progressive and integrative way throughout the program, it was not possible to assess which aspects or components, e.g., strategies or techniques, were relevant or contributed to symptom relief in our sample. This is, however, a very interesting aspect and we have added the following sentence to our discussion regarding future research:
It would also be of interest to assess which aspects of treatment are relevant for children with multidimensional psychiatric symptoms, for example, which specific CBT strategies and techniques contribute to symptom relief.
Yours Sincerely,
Sarianna Barron-Linnankoski

Reviewer 2 Report
Effectiveness of group CBT on internalizing and externalizing symptoms in children with mixed psychiatric disorders
This study looked to compare the efficacy of group BCT to treatment as usual in a group of children with mixed psychiatric conditions. Theoretically given the modern dimensional conceptualisations of psychopathology, and practically, the issues with resources in mental health care, this study has useful implications. I do however, have a number of questions, comments and suggestions for the authors below. I would be willing to review this paper again.
General comments:
· I would like to commend the authors on a simple study with good design, that has clinical and theoretical utility. In particular I commend the use of dimensional measures across psychiatric diagnoses. This fits with current conceptualisations of mental illness. I shall refer to this again later in the review.
Abstract:
· I would suggest Writing GCBT in full in the abstract at first mention
· Generally good abstract
Intro:
· First sentence needs a citation
· The first paragraph of the intro is not needed. It can be removed in its entirety to help with readability.
· You mention other studies (6, 7, 17) that, by your explanation have a similar design as yours. You need to detail these further and discuss how your study adds to the literature within this context.
· You speak about no systematic review existing. However, this is not relevant to this study and should be removed.
· This rationale could also be helped by more evidence based statements around the potential usefulness of Group CBT over TAU. You may refer to the current severe lack of mental health care workers worldwide and funding.
· You do not set hypotheses, due to limited previous evidence, however you do hypothesis texting. And therefore need hypotheses, It is fine to have hypotheses without a lot of prior evidence, it can just be informed by other populations etc. It is just a prediction after all.
· Overall the intro could be structured better and follow a funnel structure. One interesting aspect of your study is that participants had various psychiatric diagnoses and the majority had multiple, yet they recruited the same type of treatment and symptoms were measured dimensionally. I anticipate some readers may see the lack of specificity as a weakness. However, this fits within modern conceptualisations of psychopathology. I suggest the authors dedicate a paragraph to this in their introduction.
Methods
2.1:
“Children whose severity of psychiatric symptoms required individual psychotherapy were not referred to 130 GCBT (e.g., acute suicidality or psychosis)”
· Does this mean your groups where biased? That those outliners with particularly poor mental health were not within your Group program but were measured as part of the TAU group? If so this is a severe limitation and should be described further.
· Your set up of your methods is confusing. It seemed from the text you were directly comparing TAU and G-CBT. But figure 1 does not show this and instead shows group two receiving both? Please clarify.
· You provide diagnosis not the measure you used to diagnose. E.g. the MINI. Please clarify.
· Table 1 is hard to grasp with the acronyms. E.g., CBCL_Int T, M (SD). What is this? It is not in the note.
· “The mean duration of treatment 150 for the GCBT condition was 5.8 months (SD = 1.2) and for the TAU condition 3.2 months 151 (SD = 1.4)”
Can you statistically compare the main factors e.g., treatment time between the groups? Also this is a severe limitation. Is GCBT is worth it if it takes longer? Is it more cost effective actually? You need to provide so info around this.
2.5.
· How did you deal with the missing data?
· You say the distributions were non normal. This is good. But it also needs a sentence somewhere to say the analyses techniques you used are robust to mild violations in non-normality (if this is the case).
· “In all testing, p-values < 0.05 were considered statistically significant.”
Why did you not use a Bonferroni correction to correct for multiple comparisons? How many tests did you do in total? It seems to be a very large number.
Discussion
· Good work pointing out the changes in teachers. This is great information, and a important limitation to note. It would be good to point towards inter-rater reliability of this measure here to demonstrate the magnitude with wish this may be a limitation.
· Again, as with the intro. It would be good to point out the dimensional symptom assessment and mixed psychiatric conditions in this study being in line with modern conceptualisations of psychopathology, e.g., Haywood’s work and Kotov’s work.
· It would be good to give an indication of the cost saving. The other point to consider is the differences in staff resources required between the groups. If fewer staff response is required for the group sessions, this may help alleviate the issues with staffing in mental health services globally.
Author Response
Dear Editor and Reviewers: Thank you for your interest and for your comments concerning our manuscript entitled “Effectiveness of group CBT on internalizing and externalizing symptoms in children with mixed psychiatric disorders”. Your valuable comments have been of great help to us in revising and improving our paper. We have studied your comments carefully and have made the requested corrections and additions which we hope will meet with your approval. We have also had the text proofread for English language and style. Please kindly view the changes in our uploaded revised manuscript.
The main modifications in the paper and the responses to the reviewers´ comments are as follows:
Response to Reviewer 2 comments:
Abstract:
Comment 1:
I would suggest Writing GCBT in full in the abstract at first mention
Generally good abstract
Response to comment 1
Thank you for your comment and commendation. We have now written GCBT in full in the abstract to define the abbreviation.
Introduction section:
Comments 2 and 3:
First sentence needs a citation
The first paragraph of the intro is not needed. It can be removed in its entirety to help with readability.
Response to Comments 2 and 3:
Thank you for your comments. We realized from your and the other reviewer's valuable comments that the first paragraph of the original introduction was not clearly outlined. The paragraph itself contained repetitions. Moreover, the same contents are demonstrated with more depth in the following paragraphs. Based on your constructive comments, we have removed the first paragraph in its entirety to help the readability of the introduction section. We believe that this will enable us to provide the reader with a more coherent understanding of the background, rationale, and aims of the study.
Comment 4:
You mention other studies (6, 7, 17) that, by your explanation have a similar design as yours. You need to detail these further and discuss how your study adds to the literature within this context.
Response to Comment 4:
Thank you for pointing out the incompleteness of our description concerning the comparison of GCBT and TAU treatments. We have now tried to further detail prior research on the comparison of GCBT and TAU by addressing it in a separate paragraph towards the end of the Introduction section (second last paragraph). We have tried to motivate how our study adds to the literature within this context by adding a literature reference related to the comparison of GCBT and TAU as well as a comment about the potential usefulness of GCBT over TAU given the current severe lack of mental health care workers worldwide and funding challenges, thanks to your insightful comment (Comment 6).
Comment 5:
You speak about no systematic review existing. However, this is not relevant to this study and should be removed.
Response to comment 5::
Thank you for this observation. We have now removed the following phrase:“ To our best knowledge, there are no systematic reviews on CBT delivered to children with heterogeneous psychiatric diagnoses exhibiting both internalizing and externalizing symptoms”.
We have further modified the paragraph (third last paragraph of the Introduction section) on CBT in the treatment of clinical child populations with a range of psychiatric symptomology based on your constructive suggestions in Comment 8.
Comment 6
This rationale could also be helped by more evidence based statements around the potential usefulness of Group CBT over TAU. You may refer to the current severe lack of mental health care workers worldwide and funding.
Response to comment 6: Thank you for your comment. Please kindly see response to comment 4
Comment 7:
You do not set hypotheses, due to limited previous evidence, however you do hypothesis texting. And therefore need hypotheses, It is fine to have hypotheses without a lot of prior evidence, it can just be informed by other populations etc. It is just a prediction after all.
Response to Comment 7:
Thank you for pointing out that we can set hypotheses, although we did not find prior research comparing GCBT and TAU regarding externalizing symptoms. We have now added a hypothesis regarding GCBT and TAU at the end of the Introduction section.
Comment 8:
Overall the intro could be structured better and follow a funnel structure. One interesting aspect of your study is that participants had various psychiatric diagnoses and the majority had multiple, yet they recruited the same type of treatment and symptoms were measured dimensionally. I anticipate some readers may see the lack of specificity as a weakness. However, this fits within modern conceptualisations of psychopathology. I suggest the authors dedicate a paragraph to this in their introduction.
Response to comment 8:
We are very thankful for your constructive suggestions regarding better structuring the Introduction section. We have now tried to modify it towards a funnel structure.
We have edited the paragraph on CBT in the treatment of children with a range of psychiatric disorders exhibiting both internalizing and externalizing symptoms (see also response to Comment 5). We have added a description regarding the conceptualization of psychopathology towards a more dimensional approach. This new framework has also been discussed as a strength in the Discussion section.
Methods
2.1:
Comment 9 “Children whose severity of psychiatric symptoms required individual psychotherapy were not referred to 130 GCBT (e.g., acute suicidality or psychosis)”
Response to comment 9:
We noted from your observant comment that we had erroneously described the referral process for GCBT for patients with extremely severe symptoms such as e.g., acute suicidality. These patients were treated in inpatient clinics and therefore did not take part in our study, which covered all children receiving GCBT as outpatients. The Friends intervention was also administered in inpatient clinics, but due to the inpatient clinic practices, protocols were adapted to better suit the setting.
We have now tried to clarify this issue in the first paragraph of the 2.1. Participants section.
Comment 10:
Does this mean your groups where biased? That those outliners with particularly poor mental health were not within your Group program but were measured as part of the TAU group? If so this is a severe limitation and should be described further.
Response to comment 10:
Thank you for pointing out that this aspect was not clearly explained in our paper. Children treated in inpatient clinics did not participate in our study in either the TAU or GCBT condition. However, as you suggest, the lack of participants with particularly poor mental health may decrease the generalizability of our results. However, the overall symptom relief of the sample from the clinical to normal levels, based on parent-rated total problem scores, suggests that treatment is also beneficial in severe symptomology.
Comment 11: Your set up of your methods is confusing. It seemed from the text you were directly comparing TAU and G-CBT. But figure 1 does not show this and instead shows group two receiving both? Please clarify.
Response to comment 11:
Thank you for asking us to clarify our set-up regarding the TAU and GCBT treatments. We have now tried to clarify the set-up in the text (e.g., 2.2. Procedure paragraph, 2.4 Treatment paragraph), Figure 1, and the tables by defining in more detail what treatments are referred to at the different time points.
The TAU+GCBT group first received TAU followed by GCBT. The GCBT group received GCBT immediately. We have assessed symptom relief in both treatment conditions, TAU and GCBT. With regard to GCBT, we have assessed its effectiveness in both groups separately and also by combining the groups. Further, we compared the effectiveness of the TAU and GCBT conditions. We hope that our changes will help the reader to better understand the set-up.
Comment 12: You provide diagnosis not the measure you used to diagnose. E.g. the MINI. Please clarify.
Response to comment 12: Thank you for pointing out this shortcoming in our text. We have now added descriptions of the diagnostic processes at the end of the 2.3 Measures paragraph.
Comment 13: Table 1 is hard to grasp with the acronyms. E.g., CBCL_Int T, M (SD). What is this? It is not in the note.
Response to comment 13: Thank you for your comment. We have now added definitions of the abbreviations in the note section of Table 1.
Comment 14:
“The mean duration of treatment 150 for the GCBT condition was 5.8 months (SD = 1.2) and for the TAU condition 3.2 months 151 (SD = 1.4)”
Can you statistically compare the main factors e.g., treatment time between the groups? Also this is a severe limitation. Is GCBT is worth it if it takes longer? Is it more cost effective actually? You need to provide so info around this.
Response to comment 14:
We realized that we had accidentally determined the length of GBCT incorrectly based on the date that the questionnaires had been returned at post-GCBT. We are very sorry for this error.
The GCBT treatment consisted of 12 sessions, which could have been completed in close to three months if the FRIENDS program had not required a one-month break before Booster session 1 and between Booster sessions 1 and 2. This duration from Group session 1 to Booster session 2 (duration of the 12 sessions altogether) had a mean length of 4.7 months, SD: 1.0 months. We performed a t-test for the Group sessions alone (estimated mean 3 months) and in this case the result was non-significant compared to TAU (t(101) = 1.03, p = .31).
We have corrected the duration of GCBT and tried to clarify the amount of GCBT sessions regarding its duration in the 2.2 Procedure section.
We hope this is helpful for the reader to grasp the content of the treatments.
2.5.
Comment 15: How did you deal with the missing data?
Response to comment 15:
Thank you for your comment with regard to missingness. We have now clarified in the 2.5 Statistical Analyses section that no imputations were made because missingness was assumed to be completely at random.
Comment 16: You say the distributions were non normal. This is good. But it also needs a sentence somewhere to say the analyses techniques you used are robust to mild violations in non-normality (if this is the case).
Response to comment 16:
Thank you for your comment. We have now added the following sentence “The analysis techniques used are robust to mild violations in non-normality” regarding the use of the linear mixed model. Subsequent analyses regarding group comparisons were performed using non-parametric methods.
Comment 17
“In all testing, p-values < 0.05 were considered statistically significant.”
Why did you not use a Bonferroni correction to correct for multiple comparisons? How many tests did you do in total? It seems to be a very large number.
Response to comment 17:
Thank-you for your comment.
- The analysis of change over time with the whole population (mixed model) consisted of basically 6 tests, for which significance was found with regard to the CBCL parent-ratings. Significance was at the level that can withstand correction.
The subsequent analyses were performed using nonparametric tests:
- Changes during the GCBT condition were analyzed groupwise and then by combining the data of both groups (GCBT group and TAU+GCBT group; combined group)
- We tested that there were no between-group differences (GCBT vs TAU+GCBT group) in CBCL internalizing, externalizing, or total problem relative change scores during GCBT (subtraction of pre-GCBT scores from post-GCBT scores divided by pre-GCBT scores).
- We checked that the number of parent sessions attended was not associated with relative change in CBCL scores, as there was a between-group difference in the number of attended parent sessions.
- Changes in symptom ratings during follow-up were examined – comparisons for pre-GCBT to follow-up as well as post-GCBT to follow-up
- Changes in symptom ratings between the GCBT and TAU conditions were examined.
Non-parametric between-group comparisons of GCBT and TAU were nonsignificant, so there was no need for correction.
Further, we did not use Bonferroni corrections as we were testing multiple different hypothesis instead of testing the same hypotheses multiple times. We are aware of their large amount and that the results should be interpreted with caution. We have now added the following sentence into Discussion section:
Results should be interpreted with caution due to the testing of multiple scores.
Please let us know if further specification is required.
Discussion
Comment 18:
Good work pointing out the changes in teachers. This is great information, and a important limitation to note. It would be good to point towards inter-rater reliability of this measure here to demonstrate the magnitude with wish this may be a limitation.
Response to comment 18:
Thank you for your commendation and comment. Unfortunately, it was not able to measure inter-rater reliability as the TRF questionnaires were at four different time points often by a different teacher. We have, however, added a reference regarding inter-teacher reliability in the Discussion section as follows:
“Although we could not measure inter-teacher reliability because the TRF questionnaires were completed at four different time points, correlations of ratings between pairs of teachers tend to be large [50]. While changes in informants pose problems for the interpretation of results, it can be hard to avoid in a study that spans over grade levels.”
Comment 19:
Comment: Again, as with the intro. It would be good to point out the dimensional symptom assessment and mixed psychiatric conditions in this study being in line with modern conceptualisations of psychopathology, e.g., Haywood’s work and Kotov’s work.
Response to comment 19:
Thank you for your comment and for commending our study on dimensional symptom assessment in children with mixed psychiatric conditions. In addition to referring to Haywood´s and Kotov´s work regarding the modern conceptualization of psychopathology at the end of the Introduction section, we have also mentioned our study´s dimensional approach as a strength in the Discussion section, citing the work of Kotov.
Comment 20:
It would be good to give an indication of the cost saving. The other point to consider is the differences in staff resources required between the groups. If fewer staff response is required for the group sessions, this may help alleviate the issues with staffing in mental health services globally.
Response to comment 20:
Thank you for your comment. We have added indications related to cost saving regarding GCBT requiring fewer staff resources per child and being typically less expensive than individual treatments based on your helpful points.
Yours Sincerely,
Sarianna Barron-Linnankoski

Round 2
Reviewer 1 Report
I think that the major issue was fixed by the authors. Nevertheless, I would suggest revising the English and also some logical steps of the introduction are still not clear. Furthermore, I would suggest to the authors ameliorate figure 1 in the graphic and also add a legend.
Finally, another minor suggestion is to deepen the discussion with the clinical implication of their findings.
Reviewer 2 Report
The authors have addressed my main concerns.
I recommend acceptance if the authors give a clear statement within the manuscript stating why they did not correct the alpha level for multiple comparisons (with appropriate citations).
